# Sulfated Glycosaminoglycans as Viral Decoy Receptors for Human Adenovirus Type 37

**DOI:** 10.3390/v11030247

**Published:** 2019-03-12

**Authors:** Naresh Chandra, Yan Liu, Jing-Xia Liu, Lars Frängsmyr, Nian Wu, Lisete M Silva, Mona Lindström, Wengang Chai, Fatima Pedrosa Domellöf, Ten Feizi, Niklas Arnberg

**Affiliations:** 1Section of Virology, Department of Clinical Microbiology, Umeå University, SE-90185 Umeå, Sweden; naresh.chandra@umu.se (N.C.); lars.frangsmyr@umu.se (L.F.); 2Glycosciences Laboratory, Faculty of Medicine, Imperial College of London, Hammersmith Campus, London W12 0NN, UK; yan.liu2@imperial.ac.uk (Y.L.); n.wu14@imperial.ac.uk (N.W.); l.machado-e-silva@imperial.ac.uk (L.M.S.); w.chai@imperial.ac.uk (W.C.); t.feizi@imperial.ac.uk (T.F.); 3Department of Integrative Medical Biology, Umeå University, SE-90185 Umeå, Sweden; jingxia.liu@umu.se (J.-X.L.); mona.lindstrom@umu.se (M.L.); fatima.pedrosa-domellof@umu.se (F.P.D.); 4Department of Clinical Science, Ophthalmology, Umeå University, SE-90185 Umeå, Sweden

**Keywords:** adenovirus, glycosaminoglycan, cellular receptor, decoy receptor, tropism, epidemic keratoconjunctivitis, antiviral drugs

## Abstract

Glycans on plasma membranes and in secretions play important roles in infection by many viruses. Species D human adenovirus type 37 (HAdV-D37) is a major cause of epidemic keratoconjunctivitis (EKC) and infects target cells by interacting with sialic acid (SA)-containing glycans via the fiber knob domain of the viral fiber protein. HAdV-D37 also interacts with sulfated glycosaminoglycans (GAGs), but the outcome of this interaction remains unknown. Here, we investigated the molecular requirements of HAdV-D37 fiber knob:GAG interactions using a GAG microarray and demonstrated that fiber knob interacts with a broad range of sulfated GAGs. These interactions were corroborated in cell-based assays and by surface plasmon resonance analysis. Removal of heparan sulfate (HS) and sulfate groups from human corneal epithelial (HCE) cells by heparinase III and sodium chlorate treatments, respectively, reduced HAdV-D37 binding to cells. Remarkably, removal of HS by heparinase III enhanced the virus infection. Our results suggest that interaction of HAdV-D37 with sulfated GAGs in secretions and on plasma membranes prevents/delays the virus binding to SA-containing receptors and inhibits subsequent infection. We also found abundant HS in the basement membrane of the human corneal epithelium, which may act as a barrier to sub-epithelial infection. Collectively, our findings provide novel insights into the role of GAGs as viral decoy receptors and highlight the therapeutic potential of GAGs and/or GAG-mimetics in HAdV-D37 infection.

## 1. Introduction

Glycans are abundant and accessible components on the plasma membrane, and many pathogenic viruses have evolved to use glycans as attachment factors or entry receptors for their infection of host cells [1]. Glycans such as SAs, GAGs, and histo-blood group antigens are commonly used as cellular receptors, and they not only contribute to the infection but also determine the virus tropism [1]. Glycan-containing molecules are also secreted to the extracellular matrix (ECM), where they regulate various physiological processes such as inflammation, development, cell–cell adhesion, and signaling [2]. Additionally, glycans in secretions can also function as a barrier or as decoy receptors and prevent the interaction of invading viral pathogens with glycan-containing receptors on the plasma membrane of target cells [3,4]. A few reports have also demonstrated the decoy receptor function of plasma membrane-associated glycans [5,6]. In addition, enzymatic removal of SA and HS from the cell surface facilitates the egress of progeny influenza A virus (IAV) and herpes simplex virus-1 (HSV-1), respectively, from infected cells, which represent yet another level of decoy receptor function of glycans [7,8].

SAs and GAGs both are negatively charged glycans. SA monosaccharides contain a carboxylic acid group and are located at the terminus of glycan chains [9]. SAs are usually linked to galactose residues via α2,3- or α2,6-glycosidic bonds. More than 50 types of SAs have been identified so far, of which *N*-acetyl neuraminic acid is the most common type in humans. GAGs are linear polysaccharides [10]. With the exception of keratan sulfate (KS), GAGs contain repeated disaccharide units of an amino sugar (*N*-acetyl glucosamine or *N*-acetylgalactosamine) and a uronic acid (iduronic or glucuronic acid). KS is composed of repeated disaccharide units of galactose and *N*-acetyl glucosamine. The monosaccharides on GAG chains except for hyaluronic acid (HA) contain sulfate groups at their different carbon positions [10]. The disaccharide composition and sulfation level define the type of GAG; this includes heparin, HS, chondroitin sulfate (CS) -A, -B and -C (CS-B; also known as dermatan sulfate), and KS. GAG-containing proteins are referred to as proteoglycans (PGs) and mostly expressed on plasma membranes, in secretions, and in the ECM [11]. Heparin is present within mast cells and released upon cell degranulation, and is used as a model of highly sulfated HS to study HS-protein interactions [11,12]. HA, on the other hand, is a non-sulfated GAG and the only GAG that is not attached to proteins [10,11]. In the eye, SA-containing mucins are common on epithelial cells and in mucosal secretions, including tear fluid [13], whereas GAG-containing molecules such as heparan sulfated proteoglycans (HSPGs) and keratan sulfated proteoglycans (KSPGs) are present on corneal (HSPGs) and stromal keratocytes (KSPGs), respectively and in the corresponding ECM [14,15,16,17].

Human adenovirus type 37 (HAdV-D37) is one of several HAdV types that cause EKC [18]. EKC is a severe ocular infection for which there are no antiviral drugs or vaccines available [19]. A hallmark of EKC is the pronounced inflammation of the cornea and the formation of opacities that may last and affect vision for months or even years [19]. HAdVs bind to cellular receptors via the knob domain of the viral fiber protein [20]. Glycan array analysis and subsequent functional studies have revealed that HAdV-D37 fiber knob binds to a branched, SA-containing hexasaccharide corresponding to the glycan of the GD1a-ganglioside and that such glycans are required for infection of human corneal epithelial (HCE) cells [21]. Several studies have found that certain HAdVs can also interact with sulfated GAGs and that such interactions facilitate the virus infection [22,23,24]. HAdV-D37 also interacts with GAGs [21,25], but the function(s) of this interaction is unknown. In the present study, we investigated the outcome of the interaction of HAdV-D37 with GAGs in vitro and found that both soluble and plasma membrane-bound sulfated GAGs act as decoy receptors for HAdV-D37. To the best of our knowledge, this is the first report in which molecules that function as decoy receptors (i.e., GAGs) turn out to be different from molecules that function as cellular receptors (i.e., SA-containing glycans).

This study provides novel insights into the mechanisms by which GAGs can have decoy receptor activity. This has a bearing on our understanding of virus tropism, and possibly also on the future development of drugs for the topical treatment of ocular virus infections.

## 2. Materials and Methods

### 2.1. Cells, Viruses, Antibodies, and Lectins

Cells: HCE cells (a gift from Dr. Araki-Sasaki) were grown in SHEM medium (1:1; DMEM (Dulbecco’s Modified Eagle Medium) and HAM’s F12 Nutrient Mixture supplemented with 20 mM 4-(2-hydroxyethyl)-1-piperazine-ethane-sulfonic-acid (HEPES), 5 μg/mL insulin, 0.5% DMSO, 0.1 μg/mL cholera toxin, 10 ng/mL hEGF, 20 U/mL penicillin + 20 μg/mL streptomycin (PEST, Invitrogen), and 10% fetal bovine serum (FBS). A549 cells (a gift from Dr. Alistair Kidd) were grown in DMEM medium supplemented with 20 mM HEPES, 20 U/mL penicillin + 20 μg/mL streptomycin (PEST, Invitrogen), and 10% FBS. Viruses: HAdV-C5 (strain 75) and HAdV-D37 (strain 1477) were produced with and without ^35^S-labeling in A549 cells as described previously [26], with some minor modifications: viruses were eluted in phosphate buffered saline (PBS) on a NAP column (from GE Healthcare) and stored (at −20 °C) in PBS containing 10 % glycerol. Antibodies: For infection experiments, serotype-specific rabbit polyclonal antisera against HAdV-C5 and HAdV-D37 (a gift from Dr. Göran Wadell) and anti-rabbit fluorescein isothiocyanate- (FITC-) conjugated antibody (from Dako Cytomation, Glostrup, Denmark) were used. For flow cytometry and immunofluorescence, the following antibodies were used: mouse monoclonal anti-heparan sulfate (anti-HS; clone F58-10E4; from Amsbio, Abingdon, UK), mouse monoclonal anti-chondroitin sulfate (anti-CS; clone CS-56; from Amsbio), mouse monoclonal anti-keratan sulfate (anti-KS; clones 5D4 and 373E3; from Amsbio and Antibody Online, Aachen, Germany respectively), anti-RGS-His (which recognizes the epitope RGSHHHH; from Qiagen, Helden, Germany), anti-GD1a (clone EM9; a gift from Dr. Hugh Willison), donkey anti-mouse Alexa Fluor 488-conjugated (from Invitrogen, Carlsblad, USA), goat anti-mouse Alexa Fluor 488 (from Molecular Probes, Inc., Eugene, USA), and FITC-conjugated streptavidin (from Dako Cytomation). For flow cytometry, biotinylated wheat germ agglutinin (WGA) lectin was purchased from Vector Laboratories (Burlingame, USA).

### 2.2. GAGs, Chemical Reagents, and Enzymes

Heparin (from porcine intestinal mucosa, product ID H3149), hyaluronic acid (HA; from *Streptococcus equi*, product ID 94137), chondroitin sulfate A (CS-A; from bovine trachea, product ID C9819), chondroitin sulfate B (CS-B; from porcine intestinal mucosa, product ID C3788), chondroitin sulfate C (CS-C; from shark cartilage, product ID C4384), sodium chlorate (product ID 244147), heparinase III (hepIII; from *Flavobacterium heparinum*, product ID H8891), neuraminidase (neu; from *Vibrio cholerae,* product ID N7885), and chondroitinase ABC (ChABC; from *Proteus vulgaris*, product ID C3667) were purchased from Sigma (St. Louis, USA). Keratan sulfate (from bovine cornea) was a gift from Prof. Robert J Linhardt. Heparin oligosaccharide 14-mer (H014) and CSC-14 mer (CS-C 14-mer; CS014) were purchased from Iduron (Macclesfield, UK). Heparin hexamer (GAG 012) and dimer (GAG 063), and GD1a-glycan were purchased from Elicityl (Crolles, France).

### 2.3. Cloning, Expression, and Purification of Fiber Knobs

Cloning, expression, and purification of fiber knobs were performed as described previously [27]. Briefly, the fiber knob genes of HAdV-C5 and HAdV-D37 were cloned into a pQE30-Xa expression vector encoding an N-terminal His-tag using restriction sites for BamHI and XmaI (Thermo Scientific, Waltham, MA, USA). GST-tagged HAdV-D37 fiber knob was produced as following; HAdV-D37 fiber knob gene was cloned into a pGEX-6P expression vector encoding an N-terminal GST-tag using restriction sites for NcoI and XhoI (Thermo Scientific). All constructs were confirmed by sequencing (Eurofins MWG Operon, Ebersberg, Germany). His-tagged and GST-tagged proteins were expressed in *Escherichia coli* (strain M15) and *Escherichia coli* (Rosetta strain), respectively. Proteins were expressed according to the protocol from Qiagen (The QIAexpressionist^TM^). Briefly, three liters of bacterial culture were incubated at 37 °C to an optical density of 0.6. The culture was then induced with freshly prepared 1 mM isopropyl β-d-1-thiogalactopyranoside (IPTG; Thermo Scientific). After 4‒5 h, the bacterial culture was pelleted and stored at −20 °C. His-tagged fiber knobs were purified with Ni-NTA agarose beads. GST-tagged fiber knobs were purified with GST-sepharose beads followed by anion exchange (Q-sepharose) chromatography.

### 2.4. GAG Microarray

GAG oligosaccharide microarray analyses were carried out using the neoglycolipid- (NGL-) based microarray system [28]. The list of 15 GAG NGL probes is in Appendix A. Details of their preparation and the generation of the microarrays are in the Supplementary Glycan Microarray Document (Appendix A) in accordance with the MIRAGE (Minimum Information Required for A Glycomics Experiment) guidelines for reporting of glycan microarray-based data [29]. Microarray analyses of His-tagged HAdV-D37 fiber knob protein were performed essentially as described previously [30], after precomplexation with mouse monoclonal anti-poly-histidine (Ab1) and biotinylated anti-mouse IgG antibodies (Ab2) (both from Sigma) in a ratio of 4:2:1 (by weight). The protein-antibody pre-complexes were prepared by preincubating Ab1 with Ab2 for 15 min at ambient temperature, followed by addition of HAdV-D37 fiber knob and incubation for a further 15 min on ice. The protein-antibody complexes were diluted in 10 mM HEPES (pH 7.4), 150 mM NaCl, 0.02% (*v*/*v*) Blocker Casein (Pierce), 1% (*w*/*v*) bovine serum albumin (BSA, from Sigma) and 5 mM CaCl_2_ to give a final HAdV-D37 fiber knob concentration of 150 μg/mL, and overlaid onto the arrays at 20 °C for 2 h. Binding was detected with Alexa Fluor 647-labeled streptavidin (Molecular Probes). Imaging and data analysis are described in the MIRAGE complied document (Appendix A).

### 2.5. Surface Plasmon Resonance (SPR)

SPR was performed in a Biacore T200 instrument (GE, Chicago, USA). Recombinant GST-tagged HAdV-D37 fiber knobs were coupled to the CM5 sensor chip by using the amine coupling reaction according to the manufacturer’s instructions, resulting in an immobilization density of 7500–10,000 RU. The surface of the upstream flow cell was subjected to recombinant GST-coupling reaction protein and used as a reference. All binding assays were carried out at 25 °C. PBS-Tween was used as a running buffer. Analytes (GAGs) were serially diluted in running buffer to prepare a two-fold concentration series ranging from 1,000 to 3.90625 µM and then injected in series over the reference (GST) and experimental biosensor surfaces (GST-tagged HAdV-D37 fiber knob) for 60 s with a dissociation time of 60 s. Blank samples containing only running buffer were also injected under the same conditions, to allow for double referencing. The binding affinities (KDs) were calculated using BIAcore T200 evaluation software.

### 2.6. Fiber Knob Cell-Binding Assays

HCE cells were detached with pre-warmed PBS containing 0.05% EDTA, counted, and then reactivated in 10% growth medium for 1 h at 37 °C (in suspension). After 1 h, the cells were pelleted in a V-bottom 96-well plate (2 × 10^5^ cells/well) and washed once with binding buffer (BB: DMEM supplemented with 20 mM HEPES, 20 U/mL penicillin with 20 g/mL streptomycin, and 1% BSA). The cells were then incubated with HAdV-C5 fiber knobs and HAdV-D37 fiber knobs (10 μg/mL) in 100 μL BB for 1 h at 4 °C on ice. Unbound fiber knob were washed away with BB. The cells were then incubated with monoclonal mouse anti-RGS-His antibody (diluted 1:200 in BB) for 1 h at 4 °C on ice. After 1 h of incubation, the cells were washed once with BB and incubated with monoclonal donkey anti-mouse IgG Alexa Fluor 488 antibody (diluted 1:1000 in BB) for 1 h at 4 °C on ice. Thereafter, the cells were washed with FACS buffer (PBS with 2% FBS) and analyzed by flow cytometry using a FACS LSRII instrument (Becton Dickinson, Franklin Lakes, USA). The results were analyzed using FACSDiva software (Becton Dickinson, Franklin Lakes, USA). The experiment was performed with the following additions/variations: (i) HAdV-D37 fiber knobs were pre-incubated with or without increasing concentrations of heparin and HA, for 1 h at 4 °C on ice before incubation with cells, (ii) HAdV-C5 fiber knobs were pre-incubated with or without increasing concentrations of heparin, for 1 h at 4 °C on ice before incubation with cells, (iii) cells were treated with or without hepIII (1 U/mL), ChABC (0.5 U/mL), and neu (20 mU/mL) for 1 h at 37 °C before incubation with fiber knobs.

### 2.7. Virus-Cell Binding Assays

HCE cells were detached with pre-warmed PBS containing 0.05% EDTA, counted, and reactivated in 10% growth medium for 1 h at 37 °C (in suspension). The cells were then pelleted in V-bottom 96-well plates (1 × 10^5^ cells/well) and washed once with BB. ^35^S-labeled HAdV-C5 (10,000 virus particles (vp)/cell) and HAdV-D37 (10,000 vp/cell) diluted in BB were added to the cells and incubated for 1 h at 4 °C on ice. To remove unbound viruses, the cells were washed twice with PBS. Cell-associated radioactivity was measured using a Wallac 1409 liquid scintillation counter (Perkin-Elmer, Waltham, USA). The experiment was performed with the following additions/variations: (i) ^35^S-labeled HAdV-D37 viruses were pre-incubated with or without increasing concentrations of GAGs and GD1a glycan for 1 h at 4 °C on ice before adding to the cells, (ii) ^35^S-labeled HAdV-D37 viruses were pre-incubated with and without heparin (0.4 mM) and HA (0.4 mM) for 1 h at 4 °C on ice before adding to cells, (iii) ^35^S-labeled HAdV-C5 viruses were pre-incubated with and without heparin (0.4 mM) for 1 h at 4 °C on ice before adding to cells, (iv) cells were treated with or without hepIII (1 U/mL), neu (20 mU/mL), ChABC (0.5 U/mL), or a combination of enzymes for 1 h at 37 °C before incubation with viruses, (v) cells were grown in the presence or absence of 25 mM sodium chlorate before incubating with viruses. All glycans were dissolved and diluted in BB. Sodium chlorate (25 mM) was also present in the medium during reactivation of cells.

### 2.8. Analysis of Expression of HS, CS, and SA by Flow Cytometry

Experiments in which cells were treated with enzymes or grown in the presence of sodium chlorate, the expression of HS, CS, and SA on treated cells was analyzed simultaneously. As described in the previous section, after reactivation, cells treated with and without enzymes and grown with and without the presence of sodium chlorate were washed twice with BB and incubated with mouse monoclonal IgM anti-HS (diluted 1:1000), anti-CS (diluted 1:500), and mouse monoclonal anti-GD-1a (diluted 1:1000) antibodies and biotinylated WGA (1 μg/100 μL) for 1 h at 4 °C on ice. The cells were then washed with BB and incubated with secondary antibody: monoclonal donkey anti-mouse IgG Alexa Fluor 488 antibody (diluted 1:1000, against anti-HS, anti-CS, and anti-GD1a antibodies) and FITC-labeled streptavidin (diluted 1:1000, against WGA) for 1 h at 4 °C on ice. The cells were washed with FACS buffer and analyzed with flow cytometry as described above. All antibodies and WGA were diluted in BB.

### 2.9. Infection Assays

HCE cells were grown as monolayers in transparent flat-bottom 96-well plates (30,000 cells/well). The cells were then washed three times with serum-free growth medium. Viruses were added to the cells and incubated for 1 h on ice. To remove unbound viruses, the cells were washed three times with serum-free growth medium. The cells were then incubated for 44 h at 37 °C in culture medium containing 1% FBS. After 44 h of incubation, the cells were washed once with PBS (pH 7.4) and fixed with ice-cold methanol. They were then stained with polyclonal rabbit anti-HAdV-C5 (1:5000) and anti-HAdV-D37 (1:150) antisera diluted in PBS, for 1 h at room temperature (RT). The cells were washed three times with PBS and incubated for 1 h at RT with FITC-conjugated swine anti-rabbit IgG antibody (diluted 1:100 in PBS). The cells were also stained for 5 min with DAPI (4′,6-diamidino-2-phenylindole; from Vector Laboratories) diluted in PBS (1:5000). After washing three times with PBS, infected cells were counted in Trophos (Luminy Biotech Enterprises, Marseille, France). The experiment was performed with the following additions/variations: (i) HAdV-D37 was pre-incubated with or without heparin (0.4 mM) and HA (0.4 mM) for 1 h on ice before adding to cells, (ii) HAdV-C5 was pre-incubated with or without heparin (0.4 mM) for 1 h on ice before adding to cells, (iii) cells were treated with or without hepIII (1 U/mL), neu (20 mU/mL), ChABC (0.5 U/mL), or a combination of enzymes for 1 h at 37 °C before incubation with viruses, (iii) cells were grown in the presence or absence of 25 mM sodium chlorate before incubating with viruses. Heparin and HA were dissolved and diluted in serum-free growth medium.

### 2.10. Immunofluorescence of Human Corneas

Human corneas were divided into two pieces and each half was mounted with OCT cryomount (HistoLab Products AB) on cardboard and quickly frozen in propane chilled in liquid nitrogen. The samples were stored at −80 °C until further sectioning. Five- to seven-micrometer sections were cut at −23 °C using a Leica CM3050 cryostat (Leica, Heidelberg, Germany), collected on Superfrost Plus slides (Thermo Fisher), and stored at −23 °C until further processing for immunofluorescence. The sections were processed for indirect immunofluorescence using primary antibodies to HS, CS, and KS. Goat anti-mouse (GAM) IgM Alexa Fluor 488 was used as secondary antibody. Briefly, the sections were brought to RT to dry for at least 15 min, fixed with 2% paraformaldehyde (PFA) for 8–10 min, and washed three times in 0.1 M PBS for 15 min. All incubation steps were performed in a humidity chamber to keep the sections from drying. The sections were incubated with 5% normal goat serum for 15 min at RT to block non-specific binding of the secondary antibody. Thereafter, the sections were incubated overnight at 4 °C with primary antibody, which was followed by incubation with secondary antibody (30 min at 37 °C). Finally, the sections were mounted using Vectashield mounting medium with DAPI for visualization of nuclei. Secondary antibody control sections were treated as described above, except that the primary antibodies were omitted. The sections were evaluated using a Nikon Eclipse E800 microscope equipped with a SPOT RT KE slider camera (Diagnostic Instruments, Inc., Sterling Heights, MI, USA). Digital images were processed further with Adobe Photoshop CS6 software (Adobe Systems, Inc., Mountain View, CA, USA).

### 2.11. Statistical Analysis

All experiments were performed two or three times with duplicate or triplicate samples. All results are expressed as standard error of the mean (SEM). Graphical and statistical analyses were performed by using GraphPad Prism version 7 for Windows (GraphPad Software). Significance was calculated using Student’s *t*-test. All *P*-values of <0.05 were considered statistically significant.

## 3. Results

### 3.1. HAdV-D37 Fiber Knob Binds to a Broad Range of Sulfated GAGs on the Microarray

To investigate HAdV-D37 fiber knob-GAG interactions in detail and identify specific HAdV-D37 fiber knob-interacting GAGs, we generated a focused GAG microarray with neoglycolipids (NGLs) of 15 size-defined oligosaccharide fractions, short (6- or 10-mer) and long (14-mer) chains, derived from different types of GAGs including HA, CS-A, CS-B, CS-C, heparin, HS, and KS. Among these, KS oligosaccharides were prepared by digestion with two KS-specific endoglycosidases, keratanases I and II. The keratanase I-derived oligosaccharides have 6-*O*-sulfated *N*-acetyl glucosamine (GlcNAc 6S) at non-reducing ends, whereas those derived from keratanase II have Galactose (Gal) at the non-reducing ends and the Gal can be sialylated or sulfated [31]. HAdV-D37 fiber knob bound to all sulfated GAG oligosaccharide probes but not to the non-sulfated HA probe (Figure 1A,B). The binding signal appears to be dependent on the amount of GAGs immobilized on the array. We also observed a stronger degree of binding to longer GAG oligosaccharides. The intensities of binding of HAdV-D37 fiber knobs to oligosaccharides of the same chain length (14-mer) were greatest for CS-B and CS-C, followed by heparin, keratanase I-digested KS, and CS-A, and weakest for keratanase II-digested KS 14-mer. We also observed relatively weak binding to the HS oligosaccharide probes, which is likely to be due to the relatively short lengths (6- and 8-mer) of these probes in the array.

### 3.2. Soluble GAGs Reduce the Binding of HAdV-D37 to HCE Cells

To investigate the effect of soluble sulfated GAGs on HAdV-D37 and identify specific HAdV-D37 interacting GAG(s) in a cellular context, we analyzed ^35^S-radiolabeled HAdV-D37, pre-incubated with increasing concentrations of sulfated GAG polysaccharides, binding to cells. To examine the specificity in a controlled setting, we also included sulfated GAG oligosaccharides of specified lengths in the assay (Figure 2A). All GAGs inhibited binding of HAdV-D37 to cells in a dose-dependent manner, and the concentrations that inhibited 50% of virus binding (IC_50_) were obtained for all GAGs except KS and CS-C polysaccharides (Figure 2B). Heparin polysaccharide, which is highly sulfated GAG and provides more binding sites for viral fiber knobs, inhibited virus binding more efficiently than heparin oligosaccharides. We also noted that different GAG polysaccharides with relatively similar molecular weights (MWs) showed different inhibitory effects; heparin was substantially more efficient than CS-A and CS-B, whereas CS-C and KS were less efficient and inhibited virus binding only up to 40% at their highest concentration i.e. 5 mM. Similarly, heparin 14-mer was more efficient than the CS-C 14-mer. The GD1a-hexasaccharide was used as a positive control and also inhibited the binding of HAdV-D37 to cells, as expected.

### 3.3. Sulfate Groups of Soluble GAGs Are Crucial for Inhibition of HAdV-D37 Binding and Infection

To validate and determine the affinities of GAG-HAdV-D37 fiber knob interactions, we performed SPR analysis of recombinant GST-tagged HAdV-D37 fiber knobs (instead of histidine-tagged fiber knob; to avoid unspecific interactions with positively charged histidines) using GAG polysaccharides that were most efficient inhibitors in the cell-binding assay (Figure 2B). Since HA (a nonsulfated GAG) did not display any binding with HAdV-D37 fiber knob in the GAG microarray, it was included as a negative control in the SPR analysis. As expected, heparin, CS-A, and CS-B polysaccharides but not HA bound to the immobilized HAdV-D37 fiber knob with similar micromolar affinities (418–509 μM) (Table 1), which largely confirmed the results from the GAG microarray analysis and cell-based assay. This result provides substantial evidence for the direct interaction between sulfated GAGs and HAdV-D37 fiber knob. It also indicates that the presence and possibly the density of sulfate groups is important in mediating the interaction. To validate this finding in the cellular context, we examined the binding of HAdV-D37 fiber knob and HAdV-D37 to and HAdV-D37 infection of HCE cells in the presence of heparin and HA. Heparin efficiently inhibited the binding of HAdV-D37 fiber knob to cells in a dose-dependent manner, whereas HA did not affect the binding of HAdV-D37 fiber knob to cells (Figure 3A). As expected, HA affected neither the binding of HAdV-D37 to nor its infection of cells, whereas the same concentration of heparin efficiently reduced both the virus binding to and infection of cells (Figure 3B,C). Collectively, the data suggest that electrostatic charge from sulfate groups (absent in HA) may be an important factor contributing to the interaction.

### 3.4. The HAdV-GAG Interaction Is Serotype-Specific

Next, we investigated whether the inhibitory effect of heparin is specific for HAdV-D37 or whether heparin could also inhibit other HAdV types. To test this, we used HAdV-C5 virus (belonging to species C) as a reference. HAdV-C5 uses the coxsackievirus and adenovirus receptor (CAR) as a primary receptor [32] but also interacts with HSPGs on A549 and HeLa cells [22,23]. In the latter case, the interaction is mediated by the fiber shaft and not by the fiber knob [33,34]. Heparin had little or no inhibitory effect on the binding of HAdV-C5 fiber knob to HCE cells, whereas the binding of HAdV-D37 fiber knob was reduced in a dose-dependent manner (Figure 4A). Furthermore, heparin inhibited both HAdV-D37 binding to (Figure 4B) and infection of (Figure 4C) cells by approximately 80% and 90%, respectively, whereas HAdV-C5 binding to and infection of cells remained unaffected. Taken together, these data suggest that soluble sulfated GAGs exert the decoy receptor function against certain, but not all HAdV types.

### 3.5. Cell Surface HS Serves as a Decoy Receptor for HAdV-D37

We have previously shown that hepIII treatment of respiratory cells (A549 cells) increases HAdV-D37 infection [25]. HepIII removes HS efficiently from the cell surface but does not affect the expression of other GAGs or SA-containing glycans. Here, we investigated the function(s) of cell membrane HS and CS on HCE cells, which represent the ocular tropism of HAdV-D37. We first analyzed whether the HAdV-D37 fiber knob binds to HCE cells pre-treated with hepIII or ChABC, given that the latter removes CS from the cell surface. HepIII treatment significantly reduced (by ~30%) binding of HAdV-D37 fiber knob to cells, whereas ChABC treatment slightly reduced (by ~10%, but not significant) HAdV-D37 fiber knob binding (Figure 5A). Neuraminidase treatment of cells, performed as a control, also reduced HAdV-D37 fiber knob binding to cells (by ~50%). We observed that the treatment of cells with these enzymes did not affect the binding of HAdV-C5 fiber knobs. The efficiencies of the enzymatic treatments were examined by flow cytometry using monoclonal antibodies that specifically recognize HS, CS, and, SA-containing GD1a-glycans (Figure 5B). In this flow cytometry experiment, we also observed relatively lower amount of CS expression as compared to HS on HCE cells. Since we did not observe any expression of KS on untreated HCE cells, the expression of KS was not analyzed after enzymatic treatments. Furthermore, treatment of cells with any of the enzymes, or combinations thereof, reduced HAdV-D37 but not HAdV-C5 binding to cells (Figure 5C). Next, we analyzed the effect of these enzymes on HAdV-C5 and HAdV-D37 infection of cells. Interestingly, we found an increased degree of HAdV-D37 infection in cells treated with hepIII (by ~200%), whereas the infection was unaltered in ChABC-treated cells (Figure 5D). As expected, pre-treatment with neuraminidase completely abolished HAdV-D37 infection. Cells treated with combinations of these enzymes either resulted in an increased HAdV-D37 infection (hepIII + ChABC) or complete loss of infection (neu + hepIII, neu + ChABC, and neu + hepIII + ChABC). Surprisingly, removal of HS, CS, and/or SA from the cell surface resulted in an increased degree of HAdV-C5 infection. Taken together, these data suggest that cell surface molecules that carry HS and to some extent CS can function as decoy receptors for HAdV-D37, and confirm that SA-containing glycans serve as functional receptors.

### 3.6. Sulfation of Cell Surface GAGs Is Essential for Decoy Receptor Activity

To further investigate whether sulfate groups on cell surface GAGs also contribute to decoy receptor activity, we cultured HCE cells for 48 h in the presence of sodium chlorate and examined virus binding and infection in these cells. Sodium chlorate specifically inhibits de novo sulfation of cell surface GAGs and other molecules by blocking the activity of the cellular enzyme adenosine triphosphate sulfurylase (ATS), but it does not affect the synthesis of other glycans, such as SA [35,36]. We also confirmed the viability of cells grown with and without sodium chlorate before the virus-cell binding assay. The viability of cells was determined by Trypan blue-based assay [37] and under the experimental conditions was greater than 90 % (data not shown). We found that HAdV-D37 binding to sodium chlorate-treated cells was reduced by about 60% as compared to that in untreated cells, whereas the binding of HAdV-C5 was unaffected (Figure 6A). The effect and specificity of sodium chlorate were analyzed by flow cytometry using antibodies that recognize HS and CS, and with WGA, that recognizes SA as well as N-acetyl glucosamine. As expected, the cells cultured in the presence of 25 mM sodium chlorate significantly reduced binding of both anti-HS (by ~75%) and anti-CS (by ~80%) antibodies but did not affect WGA binding (Figure 6B). Surprisingly, sodium chlorate reduced infection by both HAdV-C5 and HAdV-D37 (Figure 6C). This may be due to other effects of sodium chlorate on cellular processes, which may alter the infection by several HAdV types. This finding suggests that sulfate groups present on cell surface GAGs contribute to overall HAdV-D37 binding to HCE cells and the decoy receptor activity.

### 3.7. Distribution of GAGs in Human Corneal Epithelium

EKC-causing HAdVs usually replicate in the outermost layer of the cornea, i.e., the epithelium. To our knowledge, the infection of EKC-causing HAdVs in sub-epithelial layers such as the stroma has never been demonstrated [38]. We hypothesized that the distribution of GAGs in the corneal epithelium may contribute to limit the viral tropism to the epithelial layer. Biochemical extraction of proteoglycans from human corneal explants suggested that HS and KS are present in epithelial and stromal cells, respectively [15,39]. Immunohistochemical studies have confirmed the presence of KS in the stroma [16,17]. However, there are no studies (as to the best of our knowledge), addressing the spatial distribution of HS and CS in primary human corneal epithelium. This encouraged us to perform immunofluorescence to localize HS, CS as well as KS in the healthy human corneal epithelium using anti-HS, anti-CS, and anti-KS antibodies. We observed staining of HS (Figure 7A) and CS (Figure 7B) throughout the epithelium. Slightly more intense staining of CS was observed in the most superficial layer of the epithelium, whereas the antibody against HS stained all layers equally. Importantly, the basement membrane was extensively stained with the anti-HS antibody (Figure 7A). This observation is in an agreement with previous reports, suggesting the presence of HSPGs in the corneal basement membrane [40,41]. No immunolabeling was detected in the control section (Figure 7C). We did not observe any staining with anti-KS antibodies, which is in an agreement with previous reports showing little or no KS in the human corneal epithelium [16,17]. This finding provides an insight into the possible decoy receptor function of HS and CS in the corneal epithelium.

## 4. Discussion

In this study, we report a novel decoy receptor function of soluble and plasma membrane-bound sulfated GAGs, which limit the infection of EKC-causing HAdV in human corneal epithelial, (HCE) cells and have the potential to also inhibit other SA-binding viruses. Glycans in secretions and on plasma membranes can determine the tropism and the fate of the infection [1,42,43]. For instance, it has been suggested that the respiratory and ocular tropism of IAVs that commonly bind to α2,6- and α2,3-linked SA, respectively, is determined by the predominance of α2,6-linked SA on respiratory cells and α2,3-linked SA in airway secretions and conversely, by the predominance of α2,3-linked SA on ocular cells and α2,6-linked SA in ocular secretions [42,44,45]. It has also been proposed that the polarity of SA-containing glycans in cells and secretions of ocular and respiratory tissues contribute to determining the ocular tropism of EKC-causing HAdVs [46]. Studies have shown that binding of invading IAV to SA-containing mucins on the mucosal-surface impedes the access of virus to functional receptors and limits subsequent infection of underlying cells [47,48,49]. This highlights the importance of mucosal mucins as decoy receptors of IAV, although, IAV evades this barrier by cleaving SA from mucins using its neuraminidase. Recently, McAuley et al. reported that cell surface SA-containing mucin glycoprotein 1 (MUC1) functions as a decoy receptor for IAV and influences the severity of infection [6]. However, the function of GAGs as decoy receptors has rarely been reported.

To our knowledge, the limiting effect of cell surface GAGs during virus infection cycle was first reported in the study by Hadigal et al., where plasma membrane HS was found to limit the egress of HS-binding HSV-1 from infected cells [8]. In the present study, we report a new finding that shows the limiting effect of both soluble and cell surface GAGs at the cellular entry step of a SA-binding virus. GAGs, and also sulfated monosaccharides such as GlcNAc-6S-, Gal-6S-, and Gal-3S-containing mucins, are abundant in airway mucus [50,51,52] but are present in low amounts in the tear fluid [53,54]. We suggest that sulfated molecules in mucosal secretions, for example, in airways function as putative decoy receptors for EKC-causing HAdVs. In support of this, we showed the ability of sulfated GAGs to inhibit HAdV binding to HCE cells. Heparin oligo- and polysaccharides were more potent inhibitors of virus binding than other GAGs of similar size. CS-C polysaccharide did not inhibit HAdV-D37 binding as efficiently as heparin, which was a surprise since HAdV-D37 fiber knob bound more efficiently to CS-C 14-mer than to heparin 14-mer in the GAG microarray. We speculate that this may be due to differences in affinity and avidity: in the GAG microarray, we analyzed interactions of individual fiber knobs, whereas in the cell-based assay we analyzed the binding of viruses, which contain 12 fibers. This difference could also be due to the different presentation of GAG moieties in the binding inhibition assay and microarray (polysaccharides in solution vs lipid-linked oligosaccharides on liposomes, respectively). This emphasizes the importance of having complementary approaches in studies of GAG interactions. KS inhibited the virus binding weakly and did not reach IC_50_ levels, which was in accord with relatively low intensity of binding to arrayed KS oligosaccharides. Altogether, this study highlights the potential decoy receptor function of soluble GAGs for other SA-binding viruses.

Attempts to analyze HAdV-D37 fiber knob interactions structurally with dimers to hexamers of heparin have been unsuccessful, resulting only in weak densities covering the positively charged, SA-binding central cavity on the top of the fiber knob. This indicated that interactions between HAdV-D37 fiber knobs and sulfated GAGs are likely to be mediated by weak charge-dependent contacts involving negatively charged sulfate groups on GAG molecules and the positively charged HAdV-D37 fiber knob (isoelectric point (pI): 9.14) [25]. Therefore, densely presented sulfate groups on GAG chains may allow more contacts and lead to stronger interactions. In agreement with this, we recently found that transient charge-dependent contacts contribute to the interaction between a positively charged steering rim in the fiber knob of HAdV-C52 and negatively charged polysialic acid [55]. We suggest that the overall positive charge on the fiber knob contributes to this effect, thus, a similar effect can be expected for other HAdVs that contain a positively charged fiber knob. Previously, amino acid sequence analysis of some EKC-causing HAdVs revealed that their fiber knobs contain an overall positive charge [56], thus, sulfated GAGs may act as decoy receptors for these viruses. The pI of the HAdV-C5 fiber knob is lower (pI: 5.67), which may explain the negligible effect of heparin on HAdV-C5 fiber knob and HAdV-C5 binding to and HAdV-C5 infection of cells. Our finding is in contrast to previous reports suggesting that cellular HSPGs are important for HAdV-C5 infection [22,23]. These latter findings were obtained using HeLa and A549 cells. The relative expression of GAGs and other receptors may vary in different cell lines, which may explain the different results.

Arnberg et al, have previously suggested that the cell surface HS inhibits the entry of HAdV-D37 into A549 cells [25]. In the light of the current study, we re-interpret this as HS traps HAdV-D37 on the cell surface and prevents virus binding to functional, SA-containing receptors. Conversely, hepIII removes such traps and facilitates the infection. In the present study, we followed up on these results using physiologically relevant cells, i.e., HCE cells. We demonstrate that HAdV-D37 (via its fiber knob) not only interacts with soluble sulfated GAGs but also HS on the cell surface. ChABC treatment also reduced HAdV-D37 binding to cells, which may be a consequence of disrupted avidity-dependent interactions of HAdV-D37 with CS. Interestingly, hepIII treatment (but not ChABC treatment) substantially enhanced infection of cells by HAdV-D37, which suggest that HS presented by HCE cells (but not CS), function as decoy receptors for HAdV-D37. These results support that HS function as a trap that prevents/delays HAdV-D37 interactions with neighboring SA-containing glycans. Surprisingly, we also noted that removal of HS, CS, and SA increased HAdV-C5 infection. HAdV-C5 and other species C HAdV types are unique in that the hypervariable region loop on their hexon protein contains a large number of negatively charged residues, which probably results in an overall negative charge on the capsid of these viruses [57]. It has been suggested that this negative charge can cause charge-dependent repulsion with negatively charged glycans on the cell surface [58,59]. Therefore, the removal of negatively charged glycans from the cell surface probably reduces the charge-dependent repulsion, which allows a more efficient interaction between the virus and the cellular receptor and resulting in an increased infection. Removal of CS also significantly increased HAdV-C5 infection, albeit to a lesser extent. This may be due to relatively low amounts of CS on HCE cells, which might not exert a high enough repulsive effect.

Since hepIII treatment enhanced HAdV-D37 infection of cells, we hypothesized that inhibition of de novo sulfation on the surface of HCE cells with sodium chlorate would result in reduced virus binding and enhanced infection. As expected, sodium chlorate treatment inhibited HAdV-D37 but not HAdV-C5 binding to cells, which can be explained by reduced sulfation of HS and CS on the cell surface. Considering our hypothesis about the decoy receptor function of GAGs, it was surprising that the infection of HAdV-C5 and HAdV-D37 was substantially reduced in sodium chlorate-treated cells. Since there was no significant difference in WGA binding to treated and non-treated cells, it is unlikely that the reduced infection of HAdV-D37 was due to sodium chlorate affecting glycans other than GAGs, including SA. Moreover, even if sodium chlorate treatment had altered SA expression, such an effect would not explain the reduced infection by HAdV-C5, which uses CAR as a cellular receptor. We suggest that since sodium chlorate is a metabolic inhibitor, it may interfere with either cellular or viral factors that are crucial for HAdV virus entry or later steps in the virus infection cycle. Searching the literature, we learned that sodium chlorate treatment also affects the sulfation of other cellular factors such as tyrosines [60]. Tyrosine sulfation on membrane proteins, e.g., CCR5 and PSGL-1, has been shown to be critical for the cellular entry of HIV-1 and enterovirus-71, respectively [61,62]. Proteins with sulfated tyrosines are also engaged in intracellular transport [63,64], so the reduced virus infection observed in sodium chlorate-treated cells may be the result of hitherto unidentified downstream mechanisms that require proteins with sulfated tyrosines. 

A hallmark of HAdV-associated EKC is the accumulation of sub-epithelial infiltrates of immune cells in the corneal stroma [65]. These cells are recruited by cytokines such as IL-8, which can be produced by human corneal and conjunctival epithelial cells and stromal fibroblasts as a response to infection [66,67]. It has been shown that in vitro, viruses can penetrate HS-containing Matrigel from the apical side to infect human corneal stromal fibroblasts, resulting in recruitment of immune cells from the basolateral side [68]. However, to the best of our knowledge, it has never been demonstrated that the progeny virus can actually cross the corneal epithelium, penetrate the basement membrane, and infect stromal cells in vivo. Unfortunately, HAdV-D37 does not replicate in mouse or rabbit corneal stroma [69,70], which makes it challenging to address this question in vivo. The epithelial basement membrane appears to be an efficient barrier of several HS-binding viruses [71,72]. Our immunofluorescence data shows that both HS and CS are present throughout the corneal epithelial layer, and also that the basement membrane is rich in HS. In agreement with this, there are reports suggesting that corneal epithelial cells synthesize HSPGs [15,39]. The plasma membrane of epithelial cells mainly expresses HSPGs, e.g., syndecan 1–4 [73,74]. Epithelial cells also secrete HSPGs into the basement membrane [40,75]. We propose that GAGs present on the cell surface, in secretions, and in the basement membrane may act as decoy receptors and limit virus infection by preventing the intercellular transmission of the virus as well as the passage of virus from the epithelium to the stroma (Figure 8).

In summary, this study adds to our understanding of the potential role of GAGs in determining the fate of infection and the tropism of SA-binding viruses. This also suggests that GAG-based and/or GAG-mimetics can be used as antivirals not only against viruses that use GAGs but also SA-containing glycans as functional receptors.

## 5. Conclusions

Glycans are important molecules that are present inside and outside cells and also in secretions, where they regulate development, signaling, cell adhesion, and other processes. Due to their abundance on plasma membranes, glycans are frequently used as attachment factors or receptors by pathogenic viruses, resulting in infection of the host cell. In this study, we found that sulfated GAGs function as decoy receptors for ocular infection by HAdV-D37, a virus that normally uses SA-containing glycans as cellular receptors. Our study indicates that the structure, density, and charge of glycans in secretions and on cells, may have been evolved as a consequence of the evolutionary pressure caused by glycan-interacting viruses. We also propose that glycans, in particular, GAGs, on cells and in secretions have been underestimated as natural inhibitors of glycan-binding viruses and that these molecules should be investigated further to understand their impact on viral tropism and for antiviral drug development.

## Figures and Tables

**Figure 1 viruses-11-00247-f001:**
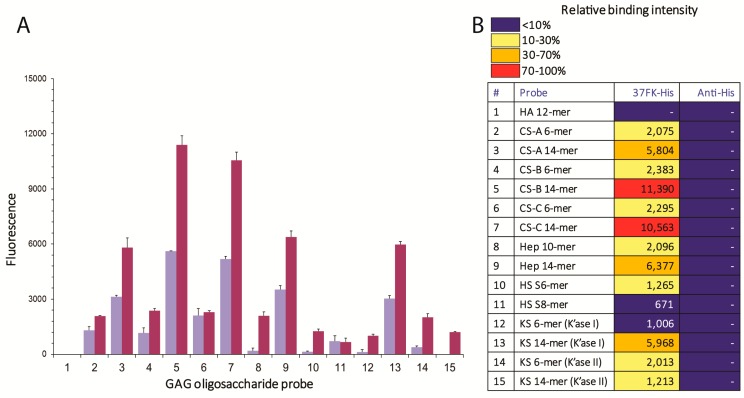
Glycan microarray analysis of HAdV-D37 fiber knob interaction with glycosaminoglycan (GAG) oligosaccharides showing selective binding to sulfated GAG oligosaccharides. (**A**) Histogram chart showing fluorescence intensities of binding of His-tagged HAdV-D37 fiber knobs as means of duplicate spots at 2 fmol/spot (purple) and 5 fmol/spot (red). Error bars represent half of the difference between the two values. (**B**) Heat map of relative intensities and fluorescence binding scores of HAdV-D37 fiber knob (means of duplicate spots at 5 fmol/spot). No binding was detected with the detection antibodies in the absence of His-tagged HAdV-D37 fiber knob. The probe list with sequences of GAG oligosaccharides is in Appendix A.

**Figure 2 viruses-11-00247-f002:**
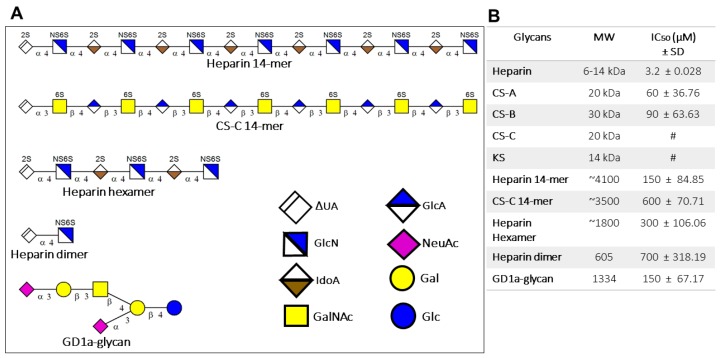
Sulfated GAGs and GD1a-glycan and their effect on HAdV-D37 binding to human corneal epithelial (HCE) cells. (**A**) Only the structures of GAG oligosaccharides are shown as representatives. GAG polysaccharides may exist as a heterogeneous mixture of GAG chains. Structure of GD1a-glycan is also shown. (**B**) Inhibitory effect of sulfated GAGs and GD1a-glycan on ^35^S-labeled HAdV-D37 virus binding to cells. The figure shows the MWs of sulfated GAGs and GD1a-glycans and their corresponding IC_50_ values. # IC_50_ level was not reached.

**Figure 3 viruses-11-00247-f003:**
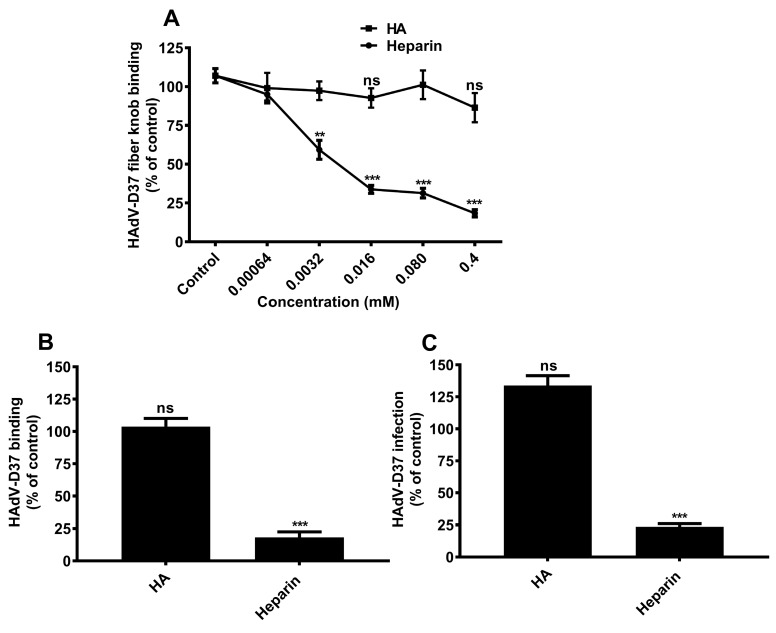
Sulfate groups of glycosaminoglycans (GAGs) are critical for interaction between HAdV-D37 and GAGs. (**A**) Binding of His-tagged HAdV-D37 fiber knobs, pre-incubated with increasing concentrations of heparin and HA, to HCE cells. Binding of fiber knobs to cells is presented as percentage binding of untreated fiber knobs. (**B**) Binding of ^35^S-labeled HAdV-D37, pre-incubated with heparin (0.4 mM) and HA (0.4 mM), to HCE cells. Binding of virus to cells is presented as percentage binding of untreated viruses. (**C**) HAdV-D37, pre-incubated with heparin (0.4 mM) and HA (0.4 mM), infection of HCE cells. Infection of cells is presented as percentage infection of untreated virus. Error bars represent mean ± SEM. ns = not significant. ** *P* < 0.01 and *** *P* < 0.001 relative to control.

**Figure 4 viruses-11-00247-f004:**
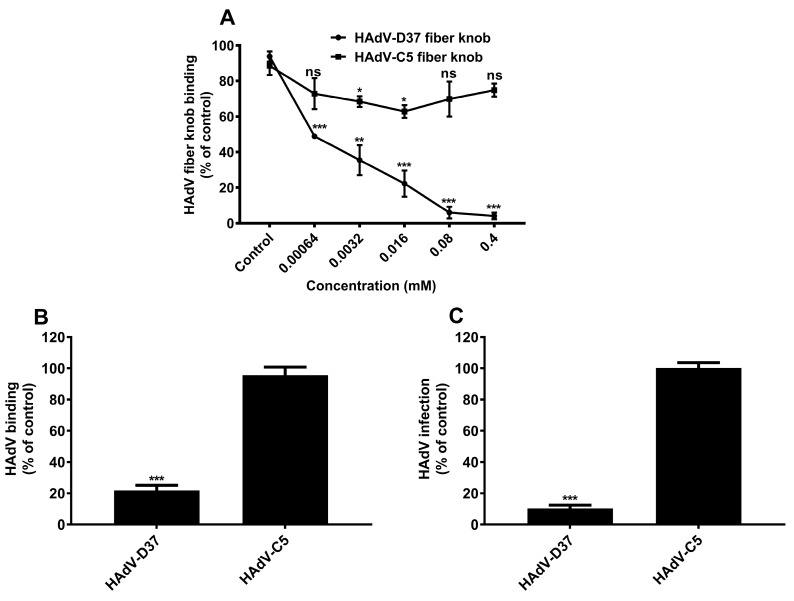
Interaction of HAdV with heparin is serotype-specific. (**A**) Binding of His-tagged HAdV-C5 fiber knobs and HAdV-D37 fiber knobs, pre-incubated with increasing concentrations of heparin, to HCE cells. Binding of fiber knobs to cells is presented as percentage binding of untreated fiber knobs. (**B**) Binding of ^35^S-labeled HAdV-C5 and HAdV-D37, pre-incubated with heparin (0.4 mM), to HCE cells. Virus binding to cells is presented as percentage binding of untreated viruses. (**C**) HAdV-C5 and HAdV-D37, pre-incubated with heparin (0.4 mM), infection of HCE cells. Virus infection of cells is presented as percentage infection of untreated viruses. Error bars represent mean ± SEM. ns = not significant. * *P* < 0.05, ** *P* < 0.01, and *** *P* < 0.001 relative to control.

**Figure 5 viruses-11-00247-f005:**
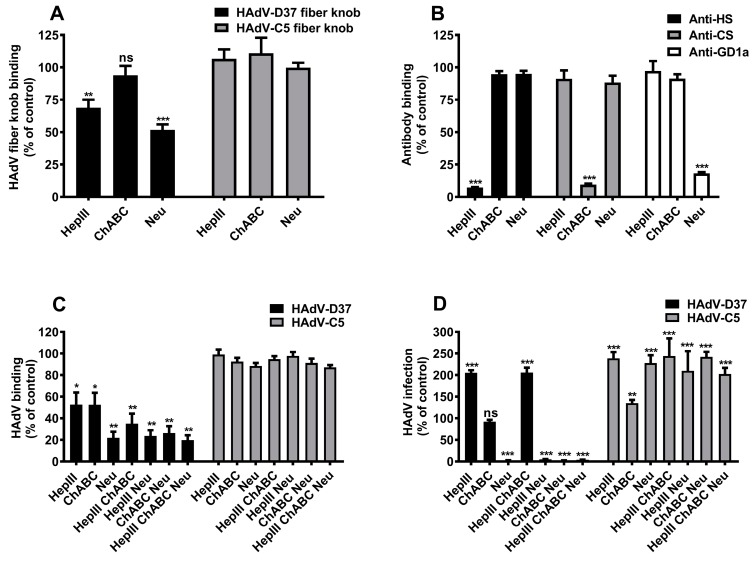
Cell surface GAGs contribute in HAdV-D37 binding to but not infection of HCE cells. (**A**) His-tagged HAdV-C5 fiber knobs and HAdV-D37 fiber knobs binding to cells pre-treated with hepIII, ChABC, and neuraminidase (neu). Binding of fiber knobs to cells is presented as percentage binding of untreated cells. (**B**) Expression of cell surface HS, CS, and SA on cells pre-treated with hepIII, ChABC, and neu. Binding of antibodies to cells is presented as percentage binding of untreated cells. (**C**) Binding of ^35^S-labeled HAdV-C5 and HAdV-D37 to cells pre-treated with hepIII, ChABC, neu, and/or combinations of enzymes. Binding of virus to cells is presented as percentage binding of untreated cells. (**D**) HAdV-C5 and HAdV-D37 infection of cells pre-treated with hepIII, ChABC, neu, and/or combinations of enzymes. Virus infection of cells is presented as percentage infection of untreated cells. Untreated cells were used as controls. Error bars represent mean ± SEM. ns = not significant. * *P* < 0.05, ** *P* < 0.01, and *** *P* < 0.001 relative to control.

**Figure 6 viruses-11-00247-f006:**
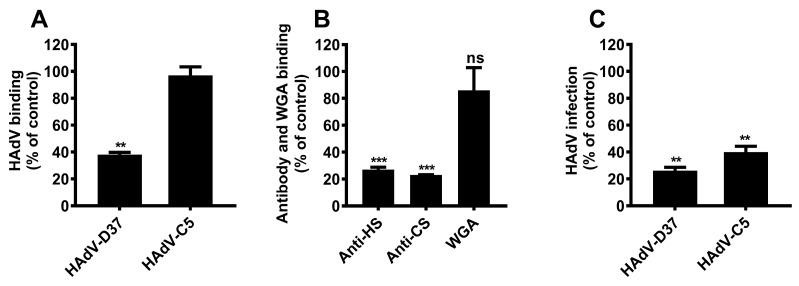
Sulfate groups on cell surface GAGs play multiple roles during HAdV binding to and infection of HCE cells. (**A**) Binding of ^35^S-labeled HAdV-C5 and HAdV-D37 to cells grown with 25 mM sodium chlorate for 48 h prior to the binding experiment. Binding of virus is presented as percentage binding of untreated cells. (**B**) Cell surface expression of HS, CS, and total SA on cells grown with 25 mM sodium chlorate. Binding of antibodies and WGA to cells is presented as percentage binding of untreated cells. (**C**) HAdV-C5 and HAdV-D37 infection of cells grown with 25 mM sodium chlorate for 48 h prior to the infection experiment. Virus infection of cells is presented as percentage infection of untreated cells. Cells grown without sodium chlorate were used as controls. Error bars represent mean ± SEM. ns = not significant. ** *P* < 0.01 and *** *P* < 0.001 relative to control.

**Figure 7 viruses-11-00247-f007:**
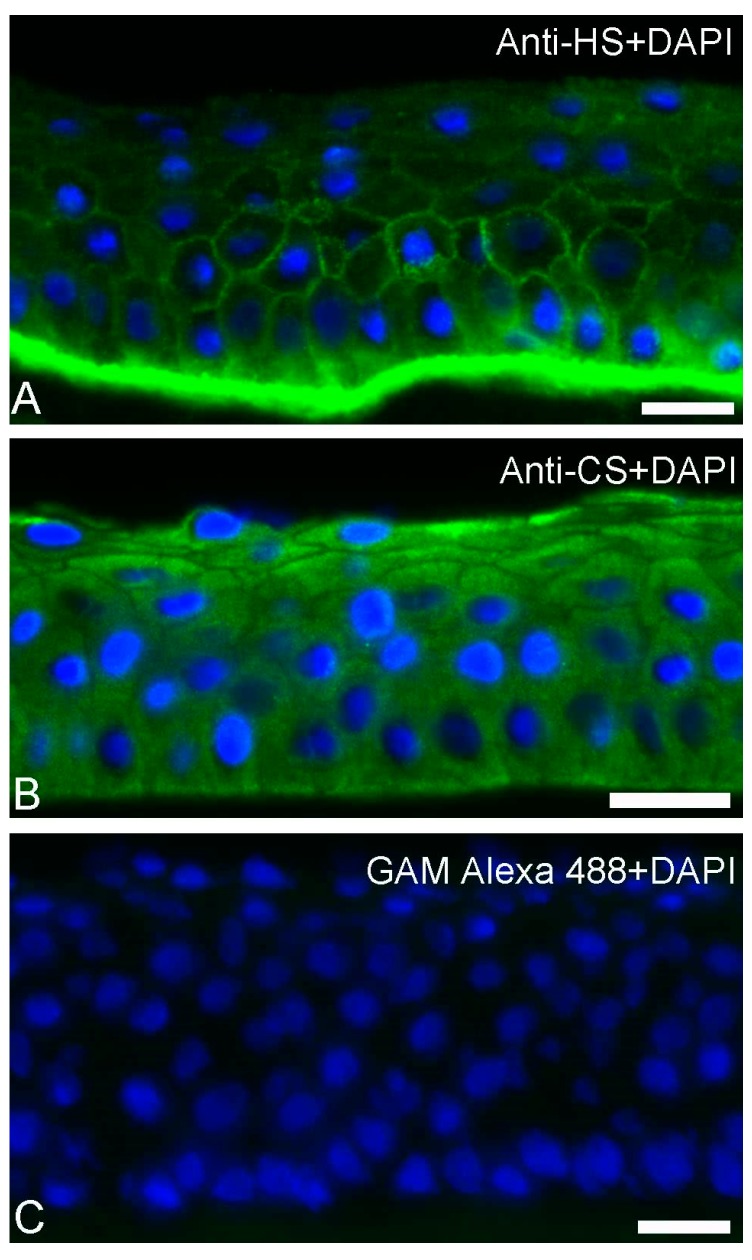
The presence of HS and CS on healthy human corneal epithelium tissue. Human corneal epithelium was labeled (green appearance) with antibodies to HS (**A**) and CS (**B**). The staining with only secondary antibody was used as a control (**C**). Nuclei were stained using DAPI (blue appearance). All bars represent 20 µM.

**Figure 8 viruses-11-00247-f008:**
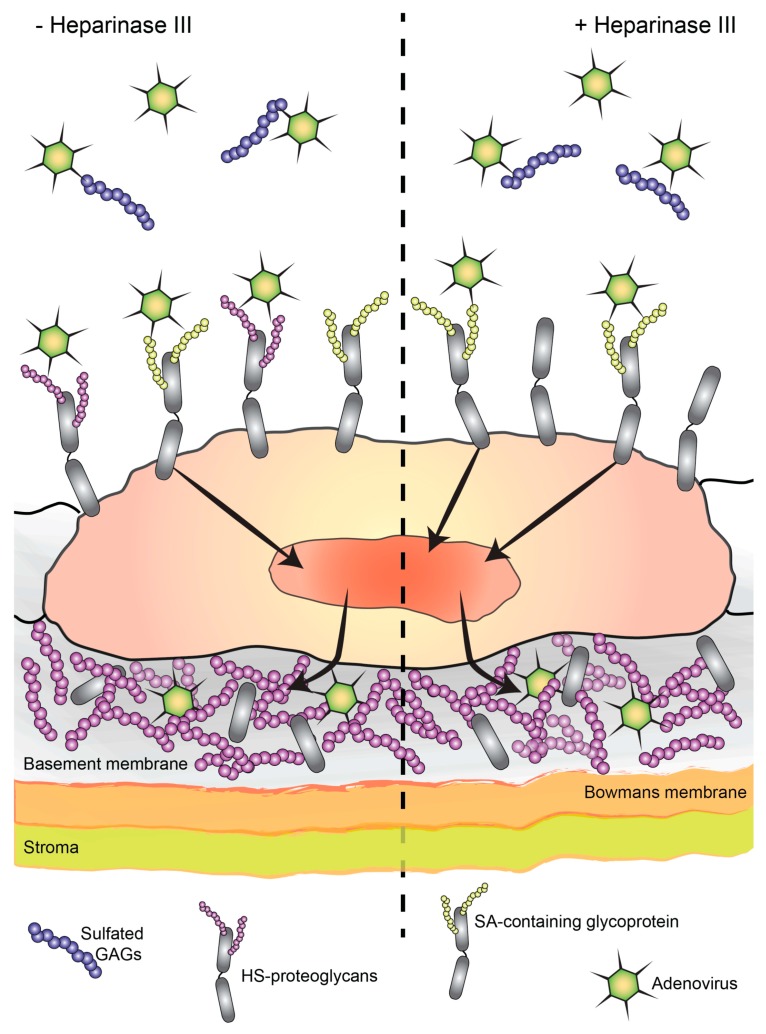
Model showing the decoy receptor function of sulfated GAGs present in secretions, on the cell surface, and in the basement membrane during HAdV-D37 infection cycle. Extracellular, soluble sulfated GAGs bind to and engage incoming viruses and the binding of viruses to their corresponding glycan receptors. On the plasma membrane, both HS- and SA-containing glycans contribute to the overall binding of HAdV-D37 to HCE cells, but only viruses bound to SA-containing glycans lead to productive infection. Although the removal of HS from the cell surface reduces overall binding, it allows more viruses to bind to SA-containing glycans, which consequently enhances virus infection. The model also demonstrates that basement membrane, which is rich in soluble HSPGs, e.g., perlecan, may restrict the passage of viruses from the corneal epithelium to the stroma.

**Table 1 viruses-11-00247-t001:** Surface plasmon resonance (SPR) analysis of the interaction of HAdV-D37 fiber knob with different GAG polysaccharides.

**Ligand** **(immobilized GST-tagged** **HAdV-D37 fiber knob)**	**Analyte (in solution)**	**MW**	**K_D_ (** **μM)** **± SD**
	Heparin	6–14 kDa	452 ± 34.29
	CS-A	20 kDa	418 ± 6.85
	CS-B	30 kDa	506 ± 50.91
	HA	10–30 kDa	ND#

# Measurable binding was not detected.

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
