# Peer review of "Sulfated Glycosaminoglycans as Viral Decoy Receptors for Human Adenovirus Type 37"

_viruses, 2019, doi:10.3390/v11030247_

Round 1

Reviewer 1 Report

The paper was well written, however, there was quite a lot of jargon to weed through and was confusing at times. While the jargon is explained in the abstract there is no definition in the text for key items such as HCE and 37FK. The jargon extended into sub headings where it would be clearer to the reader that is skimming the paper. For example, 3.1.3: FK binds to.... would be clear if stated Fiber Knobs bind to... I understand it is hard to avoid but in many cases I found myself having to keep going back and refreshing my memory on if "37FK" were just the Ad37 Fiber Knobs and not a mutated virus. Then adding in the glycan abbreviations makes some of the sentences hard for the reader to comprehend on the first read which can distract from a very good paper.

In the discussion section you have raised an interesting hypothesis, the possibility of whether mucosal secretions in airways act as decoy receptors for EKC-HAdV. Were any attempts made to obtain mucosal secretions to test this hypothesis? While the in vitro studies do support this maybe the case any attempt at showing that mucosal secretions can inhibit Ad37 infection of HCE cells would provide a stronger case for proposing this hypothesis. You state that glycans are highly abundant in the airways so it shouldn’t be difficult to see if there is at least some inhibitory factor in the secretions. Second, while many EKC strains do infect the respiratory tract is it through the respiratory tract that patients develop EKC or is does infection come from direct infection of the eye? If the latter is the main route of Ad infection leading to EKC, the presence of glycans in respiratory secretions would have little to do with preventing Ad infection leading to EKC.

You mention the ever present issue of human Ad’s not being able to infect mouse or rabbits. Do you know if EKC has ever be attempted in either the Cotton rat or Syrian hamster?

Author Response

We thank this expert for reviewing our manuscript and for providing constructive comments and suggestions. Below are our responses point by point to the specific comments.

Point 1- The paper was well written, however, there was quite a lot of jargon to weed through and was confusing at times. While the jargon is explained in the abstract there is no definition in the text for key items such as HCE and 37FK. The jargon extended into sub headings where it would be clearer to the reader that is skimming the paper. For example, 3.1.3: FK binds to.... would be clear if stated Fiber Knobs bind to... I understand it is hard to avoid but in many cases I found myself having to keep going back and refreshing my memory on if "37FK" were just the Ad37 Fiber Knobs and not a mutated virus. Then adding in the glycan abbreviations makes some of the sentences hard for the reader to comprehend on the first read which can distract from a very good paper.

Author’s response: We agree and changed the manuscript according to the suggestions. We now provide the extended name of the HCE cell line in the introduction. We also spelled out “HCE” at its first appearance in the introduction and its first appearance in the discussion. We now use “HAdV-D37 fiber knob” instead of “37FK” throughout the manuscript.

Point 2- In the discussion section you have raised an interesting hypothesis, the possibility of whether mucosal secretions in airways act as decoy receptors for EKC-HAdV. Were any attempts made to obtain mucosal secretions to test this hypothesis? While the in vitro studies do support this maybe the case any attempt at showing that mucosal secretions can inhibit Ad37 infection of HCE cells would provide a stronger case for proposing this hypothesis. You state that glycans are highly abundant in the airways so it shouldn’t be difficult to see if there is at least some inhibitory factor in the secretions. Second, while many EKC strains do infect the respiratory tract is it through the respiratory tract that patients develop EKC or is does infection come from direct infection of the eye? If the latter is the main route of Ad infection leading to EKC, the presence of glycans in respiratory secretions would have little to do with preventing Ad infection leading to EKC.

Author’s response: The reviewer raises a good point. We actually have preliminary data showing that nasal secretions, saliva, and bronchial lavage efficiently prevents HAdV-D37 binding to HCE cells, but tear fluid does not. However, these results form the basis of another manuscript and cannot be separated from that. Regarding transmission route, EKC-causing adenoviruses are very rarely isolated from the airways of patients with respiratory illness. There are a few (species B) HAdV-types such as HAdV-B3 that cause conjunctivitis (but not keratitis) and also respiratory illness. This disease is called pharyngoconjunctival fever (PCF) and should not be mixed up with EKC. The PCF-causing HAdVs use different receptors (CD46 and DSG-2 mainly). We agree with the reviewer that this question is interesting. FYI, we wrote a paper about the route of infection of sialic acid-binding viruses (including influenza A virus) in 2005: Kumlin et al Lancet Inf. Dis.

Point 3- You mention the ever present issue of human Ad’s not being able to infect mouse or rabbits. Do you know if EKC has ever be attempted in either the Cotton rat or Syrian hamster?

Author’s response: We know that other labs have made some efforts to infect e.g. rodents and rabbits with EKC-causing HAdVs, but without success: Mukherjee et. al., Invest Ophthalmol. Vis. Sci. 2015, and Romanowski et. al., Invest Ophthalmol. Vis. Sci. 1998, respectively.

Reviewer 2 Report

The submitted work by Chandra et al., investigates whether the interaction of HAdV-37 with sulfated glycosaminoglycans (GAGs) in secretions and on plasma membranes prevents or delays virus binding to sialic acid-containing receptors and inhibits consequent infection. The authors have very nicely investigated the role of GAGs as decoy receptors for virus binding, and illustrated the therapeutic potential of GAGs in HAdV-37 infection. The authors also claim that because there is abundant HS in the basement membrane of the human corneal epithelium, it may act as a decoy and help as a barrier to sub epithelial infection.

The work is very interesting, but there is need of additional work to justify the claims of the authors.

The concept that interaction of HAdV-D37 with heparin sulphate and other sulfate groups in secretions reduces the contact and binding to sialicacid glycans, which would otherwise enhance productive infection is compelling. It would be stronger however if the authors showed interactions by immunoprecipitation and then confirmation. For example, perform fiber knob immunoprecipitation by chondroitin sulfate-A, B and C, and show that it binds CS-B and C better.

The authors should consider confirming some of their data in primary corneal epithelial cells, which are not difficult to obtain. The cell type used in all their study, HCE cells are a SV40 transformed cell line, and to what degree it represents a normal corneal epithelial cell phenotype is questionable. The authors should perform at least one experiment in primary cells as a proof to show that the conclusions drawn are valid.

For section 2.8, is there a control SPR with just GST in the empty vector?

For section 3.4. The HAdV-GAG Interaction is Serotype Specific:

The authors should perform this assay in presence of ECM proteins. It is very likely that they would see differences in infectivity. Also, the authors should show this in primary epithelial cells and in A549 cells.

Results 3.7: I agree with the authors that there is no data available to show that the viruses can pass beyond epithelial basement membrane. However, the authors have no actual data for the interpretation of virus restriction from sub-epithelium and stroma.

Fig.7: It would add to the paper and our understanding to use fluorescent labeled virus to visualize binding to HS/CS.

Minor comments:

Line 156: The terminologies 5FKs and 37Fks for HAdV-C5 and -D37 fiber knobs are not generally used. Please consider using common abbreviations for the readers understanding. The overuse of abbreviations in the paper makes it hard to read.

Line 405: "presented"

Author Response

We appreciate the constructive comments/suggestions. Below are our responses point by point to the specific comments made by this reviewer. Note that the line numbers given below (author’s replies) correspond to the line numbers in the revised manuscript (word file, with track changes on).

The submitted work by Chandra et al., investigates whether the interaction of HAdV-37 with sulfated glycosaminoglycans (GAGs) in secretions and on plasma membranes prevents or delays virus binding to sialic acid-containing receptors and inhibits consequent infection. The authors have very nicely investigated the role of GAGs as decoy receptors for virus binding, and illustrated the therapeutic potential of GAGs in HAdV-37 infection. The authors also claim that because there is abundant HS in the basement membrane of the human corneal epithelium, it may act as a decoy and help as a barrier to sub epithelial infection.

The work is very interesting, but there is need of additional work to justify the claims of the authors.

Point 1: The concept that interaction of HAdV-D37 with heparin sulphate and other sulfate groups in secretions reduces the contact and binding to sialic acid glycans, which would otherwise enhance productive infection is compelling. It would be stronger however if the authors showed interactions by immunoprecipitation and then confirmation. For example, perform fiber knob immunoprecipitation by chondroitin sulfate-A, B and C, and show that it binds CS-B and C better.

Author’s response: We agree that our conclusions would be even stronger by data from the suggested experiment. However, we feel that the results obtained by GAG microarray, SPR (highly sensitive method), and cell-based assays provide sufficient evidence to justify the conclusions drawn. Moreover, our SPR data demonstrated that the interactions between HAdV-D37 fiber knobs and GAGs are of intermediate affinity, which may not be enough to pull out GAGs by immunoprecipitation. This indicates that the interaction between HAdV-D37 and sulfated GAGs is favoured by avidity-dependent mechanisms, which we also suggest in the manuscript.

Point 2: The authors should consider confirming some of their data in primary corneal epithelial cells, which are not difficult to obtain. The cell type used in all their study, HCE cells are a SV40 transformed cell line, and to what degree it represents a normal corneal epithelial cell phenotype is questionable. The authors should perform at least one experiment in primary cells as a proof to show that the conclusions drawn are valid.

Author’s response: The human corneal epithelium (HCE) cell line used in this study is actually highly similar to normal primary epithelial cells. We obtained HCE cells from Dr. Araki-Sasaki (Invest Ophthalmol. Vis. Sci. 1995), who performed extensive characterization of HCE cell line (morphological, cytological, and biochemical). They also provided evidence for epithelial features and cornea-specific markers, and showed that the cell line can differentiate in a multilayer fashion, which is rarely obtained with other transformed cell lines (which we confirmed in a previous work; Storm et. al., J. Virol. 2017). Thus, we think that the cell line represents the ocular tropism of EKC-causing HAdVs well. If the Editor requests us to perform this experiment with primary human corneal epithelial cells, we will of course consider to do so. But, if so we will need up to three more months. We have previously attempted to grow primary human corneal epithelial cells obtained from the local university hospital, but failed to grow epithelial cells to sufficient numbers. Currently, we know that we cannot access primary cells from the local hospital, so we need to either initiate an external collaboration, or purchase the cells from a commercial source (if available), and grow the cells to sufficient numbers. Thus, it is impossible for us to perform this experiment within the time frame (five days) given to resubmit the manuscript.

Point 3: For section 2.8, is there a control SPR with just GST in the empty vector?

Author’s response: In section 2.8 we only perform flow cytometry, and not SPR. We assume that the reviewer refers to section 2.5 (description of SPR). In these experiments we used GST in the reference flow cell. See line no. 161-162.

Point 4: For section 3.4. The HAdV-GAG Interaction is Serotype Specific: The authors should perform this assay in presence of ECM proteins. It is very likely that they would see differences in infectivity. Also, the authors should show this in primary epithelial cells and in A549 cells.

Author’s response: We don’t fully understand this question. We demonstrate that heparin prevented HAdV-D37 virus binding and infection, and HAdV-D37 fiber knob binding to HCE cells, but heparin did not prevent HAdV-C5. To us, these results are clear evidence of serotype specific GAG interactions. However, we have actually performed some of the experiments asked for (Storm et al, J. Virol. 2017), and as the reviewer predicted, we did observe serotype-specific effects of ECM proteins, i.e. vitronectin, fibronectin, and laminin, on HAdV infection in the absence of GAGs. Thus, in the presence of ECM proteins it would be difficult to conclude what would be the effect of GAGs and what would be the effect of ECM proteins.

Point 5: Results 3.7: I agree with the authors that there is no data available to show that the viruses can pass beyond epithelial basement membrane. However, the authors have no actual data for the interpretation of virus restriction from sub-epithelium and stroma.

Author’s response: We agree to this point. However, we think it is suitable to point out that there are no known (to us) reports about stromal infection, and the lack of animal models makes it challenging to address this question experimentally.

Point 6: Fig.7: It would add to the paper and our understanding to use fluorescent labeled virus to visualize binding to HS/CS.

Author’s response: In figure 7 we perform immunohistochemistry of human corneal epithelium, but this method is not sensitive enough to visualize co-localization of HAdV-D37 with GAGs. We have previously (Nilsson et al Nat Med 2011) used this method to show that there is abundant sialic acid-containing glycans in the human corneal epithelium. Thus, with this method it would be impossible to distinguish between HS/CS vs sialic acid-binding. There are also other molecules (e.g. integrins) in the human corneal epithelium to which HAdV-D37 also interact, which will complicate the interpretation of the outcome of such experiment.

Minor comments:

Point 1: Line 156: The terminologies 5FKs and 37Fks for HAdV-C5 and -D37 fiber knobs are not generally used. Please consider using common abbreviations for the readers understanding. The overuse of abbreviations in the paper makes it hard to read.

Author’s response: Thanks for the suggestion. We have now modified the terminology as suggested.

Point 2: Line 405: "presented"

Author’s response: Thanks for noticing the error. Corrected (line 443).

Reviewer 3 Report

The authors investigated the molecular requirements of HAdV-37 FK (37FK):GAG interactions using a GAG microarray and demonstrated that FK interacts with a broad range of sulphated GAGs. Authors removed heparin sulfate (HS) and sulfate groups from human corneal epithelial (HCE) cells by heparinase III and sodium chlorate treatments, which can reduces HAdV-37 binding to cell surface and enhanced the virus infection. Authors suggested that interaction between HAdV-37 with sulfated GAGs in sectertions and on plasma members prevents/delays the virus binding to SA receptors and inhibits subsequent infection. The new achievement provide novel insights into the role of GAGs as viral receptors.

The authors presented a big informative data which has an influence for the virus entrance to the cells from that reason I am agree for this data for the publication in Viruses.

regards

Author Response

We thank this reviewer for the very positive response.

The authors investigated the molecular requirements of HAdV-37 FK (37FK):GAG interactions using a GAG microarray and demonstrated that FK interacts with a broad range of sulphated GAGs. Authors removed heparin sulfate (HS) and sulfate groups from human corneal epithelial (HCE) cells by heparinase III and sodium chlorate treatments, which can reduces HAdV-37 binding to cell surface and enhanced the virus infection. Authors suggested that interaction between HAdV-37 with sulfated GAGs in secretions and on plasma members prevents/delays the virus binding to SA receptors and inhibits subsequent infection. The new achievement provide novel insights into the role of GAGs as viral receptors.

The authors presented a big informative data which has an influence for the virus entrance to the cells from that reason I am agree for this data for the publication in Viruses.

regards

Reviewer 4 Report

Minor Revisions

Line 17 and general: When using different Types of HAdV from different species, the species should be included in the abbreviation of the viruses, i.e. HAdV-D37 and HAdV-C5

Line 26: The authors claim, that removal of HS enhanced viral infection, however, this was only true for removal via heparinase III, and not for treatment with sodium chlorate. This should be stated here.

Line 40: Glycan-containing molecules are also secreted to the extracellular matrix (ECM)

Line 94: The authors only list a small proportion of the antibodies used within the study in the antibodies section of the materials and methods part.

Line 114: Space missing: (Eurofins MWG Operon)._His-tagged

Line 117: three liters of bacterial culture was were

Lines 121-123: The authors don’t show any data on the purity and size of the isolated proteins and can therefore skip this part of the materials and methods section.

Line 127: microarrays are in the Supplementary Glycan

Line 162: analyzed by flow cytometry using a FACS LSRII

Lines 186, 187: The degree of viable cells as determined by trypan blue is a result and therefore should be in the results part.

Line 239: The authors claim, that no immunolabeling was detected in the control sections. This is a result, and the corresponding data should be shown in the results part of the paper.

Line 249: 3.1.37. must be 3.1. 37FK

Line 311: remove the dot

Line 315/Figure 3 A: Inconsistent labeling of HAdV (Had-37FK, instead of HAdV-37FK or just 37FK)

Lines 327, 328: The authors state, that the interaction of the HAdV-C5 fiber protein with HSPGs is mediated via the fiber shaft domain and not the fiber knob, yet they use just the 5FK for their experiments.

Figures 3, 4, 5: Inconsistent Data presentation: In Figure 3, the authors show Geo mean values for 37FK binding, CPM values for HAdV-37 binding and number of HAdV-37 infected cells, in figure 4, they show values normalized to a control. In figure 5, again Geo means are shown.

Lines 353, 354: The authors state, that treatment of cells with different enzymes which remove different sulfated glycosaminoglycans does not impair 5FK binding, which is predictable, as the FK is not the domain of the HAdV-5 fiber protein that binds to HSPGs.

Lines 365, 366: Given, that the HAdV-5 fiber shaft interacts with HSPGs, which might also act as decoy receptors, it does not seem so surprisingly, that removal of these decoy receptors increases HAdV-5 infection, even though the authors do not see any effect with 5FK (which does not include the domain of the HAdV-5 fiber, which interacts with HSPGs).

Lines 394, 395: The authors claim, that sodium chlorate might affect other cellular processes, “which can also alter the infection by several HAdV types”, but fail to give a reference for this claim.

Figure 4: The authors claim that the effect is type-specific. It could also be group specific, as only two types were compared. What is with other D-types?

Figure 7: Controls not stained with primary antibodies are missing, to prove specificity of the used antibodies.

Line 431: All bars represent 20 μM.4.

Line 450: the limiting effect of cell surface GAGs during virus infection cycle was first reported in the

Discussion in general: References to the figures in the results are missing

Line 533: appears to be an efficient barrier of several HS-binding

Please revise the figures to be similar and consistent in font and size of letters

Major Revisions

Results in general: The authors use in almost every experiment His-tagged 37FK/5FK, which is not used for the Surface Plasmon Resonance experiment, “to avoid unspecific interactions with positively charged histidines” (Lines 297, 298). Why do the authors think, that this might not be also a problem for the other experiments, in which they test interactions with the same, also negatively charged sulfated glycosaminoglycans?

Author Response

We truly appreciate the constructive comments and suggestions from this reviewer, which helped us to improve the manuscript. We have adopted all the suggestions in our revised manuscript.  Please find below our point to point responses to the reviewers’ comments and suggestions. Note that the line numbers given below (author’s replies) correspond to the line numbers in the revised manuscript (word file, with track changes on).

Minor Revisions

Point 1- Line 17 and general: When using different Types of HAdV from different species, the species should be included in the abbreviation of the viruses, i.e. HAdV-D37 and HAdV-C5

Author’s response: We agree and changes have been made as suggested. 

Point 2- Line 26: The authors claim, that removal of HS enhanced viral infection, however, this was only true for removal via heparinase III, and not for treatment with sodium chlorate. This should be stated here.

Author’s response: We agree with the reviewer. We stated heparinase III in this sentence (line 27).   

Point 3- Line 40: Glycan-containing molecules are also secreted to the extracellular matrix (ECM)

Author’s response: “The” has been added (line 42)

Point 4- Line 94: The authors only list a small proportion of the antibodies used within the study in the antibodies section of the materials and methods part.

Author’s response: The detailed information of the antibodies has been added in the materials and methods section as suggested (line 97-107).

Point 5- Line 114: Space missing: (Eurofins MWG Operon)._His-tagged

Author’s response: We corrected this error (line 127).

Point 6- Line 117: three liters of bacterial culture was were

Author’s response: We corrected this grammatical error (line 130).

Point 7- Lines 121-123: The authors don’t show any data on the purity and size of the isolated proteins and can therefore skip this part of the materials and methods section.

Author’s response: Thanks for the suggestion. We removed this part.

Point 8- Line 127: microarrays are in the Supplementary Glycan

Author’s response: “The” has been added (line 140).   

Point 9- Line 162: analyzed by flow cytometry using FACS LSRII

Author’s response: “a” has been added (line 177).

Point 10- Lines 186, 187: The degree of viable cells as determined by trypan blue is a result and therefore should be in the results part.

Author’s response: We thank the reviewer for this suggestion. We moved the corresponding information in the result section, which appears in line 421-423.

Point 11- Line 239: The author’s claim that no immunolabeling was detected in the control sections. This is a result, and the corresponding data should be shown in the results part of the paper.

Author’s response: We agree with the reviewer. The corresponding sentence, which states the results in the materials and methods, has been removed. A description of these results is present in the result section (lines 461-462).

Point 12- Line 249: 3.1.37. must be 3.1. 37FK

Author’s response: Corrected (line 264).

Point 13- Line 311: remove the dot

Author’s response: Corrected (line 333).

Point 14- Line 315/Figure 3 A: Inconsistent labeling of HAdV (Had-37FK, instead of HAdV-37FK or just 37FK)

Author’s response: We thank the reviewer for noticing this error. Changes are made accordingly (Figure 3).

Point 15- Lines 327, 328: The authors state, that the interaction of the HAdV-C5 fiber protein with HSPGs is mediated via the fiber shaft domain and not the fiber knob, yet they use just the 5FK for their experiments.

Author’s response: This is correct. However, since EKC-causing HAdVs interact with sulfated GAGs through the fiber knob domain, we thought it was suitable to use the fiber knob domain of HAdV-C5 as a control. Moreover, entire fiber proteins that contain both shaft and knob are difficult to express in soluble form (for unknown reasons). We therefore used fiber knobs only.

Point 16- Figures 3, 4, 5: Inconsistent Data presentation: In Figure 3, the authors show Geo mean values for 37FK binding, CPM values for HAdV-37 binding and number of HAdV-37 infected cells, in figure 4, they show values normalized to a control. In figure 5, again Geo means are shown.

Author’s response: We agree with the reviewer. For consistency, now all data in these figures are shown as the percentage of control.

Point 17- Lines 353, 354: The author’s state, that treatment of cells with different enzymes which remove different sulfated glycosaminoglycans does not impair 5FK binding, which is predictable, as the FK is not the domain of the HAdV-5 fiber protein that binds to HSPGs.

Author’s response: This is correct. However, we decided to keep this information, since it is a suitable control.

Point 18- Lines 365, 366: Given, that the HAdV-5 fiber shaft interacts with HSPGs, which might also act as decoy receptors, it does not seem so surprisingly, that removal of these decoy receptors increases HAdV-5 infection, even though the authors do not see any effect with 5FK (which does not include the domain of the HAdV-5 fiber, which interacts with HSPGs).

Author’s response: We agree with the reviewer’s observation. Based on our result (infection data), we could also suggest GAGs as decoy receptors for HAdV-C5. But, we deliberately avoided this interpretation since HSPGs has been shown to be important for HAdV-C5 infection. In addition, the increased infection of HAdV-C5 is probably the result of reduced charge-dependent repulsion effect between GAGs and HAdV-C5 hexon protein, which is different from the decoy-receptor function of sulfated GAGs against HAdV-D37. This is discussed on lines 541-551.

Point 19- Lines 394, 395: The authors claim, that sodium chlorate might affect other cellular processes, “which can also alter the infection by several HAdV types”, but fail to give a reference for this claim.

Author’s response: This has been addressed in the discussion section: Lines 562-570 and by references 59-63 in the modified manuscript.

Point 20- Figure 4: The authors claim that the effect is type-specific. It could also be group specific, as only two types were compared. What is with other D-types?

Author’s response: We agree, and we therefore added the following sentences in discussion section (lines 519-523).

“We suggest that the overall positive charge on the fiber knob contributes to this effect, thus, a similar effect can be expected for other HAdVs that contain a positively charged fiber knob. Previously, amino acid sequence analysis of some EKC-causing HAdVs revealed that their fiber knobs also contain an overall positive charge, thus, sulfated GAGs may act as decoy receptors for these viruses”.

Point 21- Figure 7: Controls not stained with primary antibodies are missing, to prove specificity of the used antibodies.

Author’s response: We are not sure if the reviewer ask for staining of tissues that do not express GAGs? If so, we are not aware of such tissues. Or, does the reviewer wants us to use a primary antibody that does not bind to human cornea? We could probably perform such an experiment, but it may take us months to get our hands on healthy human corneal tissue and to identify such an antibody and perform the experiments asked for. But, we would prefer not to, since this would not add much or change any conclusions drawn.

Point 22- Line 431: All bars represent 20 μM.4.

Author’s response: Thanks for noticing this error. Corrected (line 470).

Point 23- Line 450: the limiting effect of cell surface GAGs during virus infection cycle was first reported in the

Author’s response: Thanks for the suggestion. The sentence has been formatted as suggested (line 489-490).

Point 24- Discussion in general: References to the figures in the results are missing.

Author’s response: We have carefully referred to each figure in the results section, but not in the discussion section. We could not see in the guidelines to authors that this is asked for, and when looking at other articles published in “Viruses”, we could not see that this is done by others. We have therefore not referred to the figures in the discussion section.

Point 25- Line 533: appears to be an efficient barrier of several HS-binding

Author’s response: Thanks for noticing this error. Corrected (line 580).

Point 26- Please revise the figures to be similar and consistent in font and size of letters

Author’s response: This has been revised as suggested.

Major Revisions

Point 1- Results in general: The authors use in almost every experiment His-tagged 37FK/5FK, which is not used for the Surface Plasmon Resonance experiment, “to avoid unspecific interactions with positively charged histidines” (Lines 297, 298). Why do the authors think, that this might not be also a problem for the other experiments, in which they test interactions with the same, also negatively charged sulfated glycosaminoglycans?

Author’s response: The SPR is the only experiment that is label free and the readout is the interaction/affinity between free fiber knobs and GAGs. Thus, in this experiment there is a risk that what we detect may be an outcome of positively charged tags interacting non-specifically with negatively charged GAGs. We could have used GST-tagged knobs in the other experiments also, but we simply did not think that this would make any difference, since in these experiments the readout is based on antibodies detecting the tags, which cannot both bind non-specifically to cells/GAGs and also be recognized by the anti-tag monoclonal antibody at the same time.